# MAGDiff: Covariate Data Set Shift Detection via Activation Graphs of Neural Networks

## Abstract

Despite their successful application to a variety of tasks, neural networks remain limited, like other machine learning methods, by their sensitivity to shifts in the data: their performance can be severely impacted by differences in distribution between the data on which they were trained and that on which they are deployed. In this article, we propose a new family of representations, called `MAGDiff`, that we extract from any given neural network classifier and that allows for efficient covariate data shift detection without the need to train a new model dedicated to this task. These representations are computed by comparing the activation graphs of the neural network for samples belonging to the training distribution and to the target distribution, and yield powerful data- and task-adapted statistics for the two-sample tests commonly used for data set shift detection. We demonstrate this empirically by measuring the statistical powers of two-sample Kolmogorov-Smirnov (KS) tests on several different data sets and shift types, and showing that our novel representations induce significant improvements over a state-of-the-art baseline relying on the network output.

## 1 Introduction

During the last decade, neural networks (NN) have become immensely popular, reaching state-of-the-art performances in a wide range of situations. Nonetheless, once deployed in real-life settings, NN can face various challenges such as being subject to adversarial attacks Huang et al. (2017), being exposed to out-of-distributions samples (samples that were not presented at training time) Hendrycks & Gimpel (2016), or more generally being exposed to a *distribution shift*: when the distribution of inputs differs from the training distribution (*e.g.*, input objects are exposed to a corruption due to deterioration of measure instruments such as cameras or sensors). Such distribution shifts are likely to degrade performances of presumably well-trained models Wiles et al. (2021b), and being able to detect such shifts is a key challenge in monitoring NN once deployed in real-life applications. Though shift detection for univariate variables is a well-studied problem, the task gets considerably harder with high-dimensional data, and seemingly reasonable methods often end up performing poorly Ramdas et al. (2014).

In this work, we introduce the Mean Activation Graph Difference (`MAGDiff`), a new approach that harnesses the powerful dimensionality reduction capacity of deep neural networks in a data- and task-adapted way. The key idea, further detailed in Section 4, is to consider the activation graphs generated by inputs as they are processed by a neural network that has already been trained for a classification task, and to compare such graphs to those associated to samples from the training distribution. The method can thus be straightforwardly added as a diagnostic tool on top of preexisting classifiers without requiring any further training ; it is easy to implement, and computationally inexpensive. As the activation graphs depend on the network weights, which in turn have been trained for the data and task at hand, one can also hope for them to capture information that is most relevant to the context. Hence, our method can easily support, and benefit from, any improvements in deep learning.

Our approach is to be compared to *Black box shift detection* (BBSD), a method introduced in Lipton et al. (2018); Rabanser et al. (2019) that shares a similar philosophy. BBSD uses the output of a trained classifier to efficiently detect various types of shifts (see also Section 4); in their experiments, BBSD generally beats other methods, the runner-up being a much more complex and

computationally costly multivariate two-sample test combining an autoencoder and the Maximum Mean Discrepancy statistic Gretton et al. (2012).

Our contributions are summarized as follows.

1. Given any neural network classifier, we introduce a new family of representations MAGDiff, that is obtained by comparing the activation graphs of samples to the mean activation graph of each class in the training set.

2. We propose to use MAGDiff as a statistic for data set shift detection. More precisely, we combine our representations with the statistical method that was proposed and applied to the Confidence Vectors (CV) of classifiers in Lipton et al. (2018), yielding a new method for shift detection.

3. We experimentally show that our shift detection method with MAGDiff outperforms the state-of-the-art BBSD with CV on a variety of datasets, covariate shift types and shift intensities, often by a wide margin. Our code is provided in the Supplementary Material and will be released publicly.

## 2 RELATED WORK

Detecting changes or outliers in data can be approached from the angle of anomaly detection, a well-studied problem Chandola et al. (2009), or out-of-distribution (OOD) sample detection Shafaei et al. (2018). Among techniques that directly frame the problem as shift detection, kernel-based methods such as Maximum Mean Discrepancy (MMD) Gretton et al. (2012); Zaremba et al. (2013) and Kernel Mean Matching (KMM) Gretton et al. (2009); Zhang et al. (2013) have proved popular, though they scale poorly with the dimensionality of the data Ramdas et al. (2014). Using classifiers to test whether samples coming from two distributions can be correctly labeled, hence whether the distributions can be distinguished, has also been attempted; see, *e.g.*, Kim et al. (2021). The specific cases of covariate shift Jang et al. (2022); Uehara et al. (2020); Rabanser et al. (2019) and label shift Storkey (2009); Lipton et al. (2018) have been further investigated, from the point of view of causality and anticausality Schölkopf et al. (2012). Moreover, earlier investigations of similar questions have arisen from the fields of economics Heckman (1977) and epidemiology Saerens et al. (2002).

Among the works cited above, Lipton et al. (2018) and Rabanser et al. (2019) are of particular interest to us. In Lipton et al. (2018), the authors detect label shifts using shifts in the distribution of the outputs of a well-trained classifier; they call this method Black Box Shift Detection (BBSD). In Rabanser et al. (2019), the authors observe that BBSD tends to generalize very well to covariate shifts, though without the theoretical guarantees it enjoys in the label shift case. Our method is partially related to BBSD. Roughly summarized, we apply similar statistical tests—combined univariate Kolmogorov-Smirnov tests—to different features—Confidence Vectors (CV) in the case of BBSD, distances to mean activation graphs (MAGDiff) in ours. Similar statistical ideas have also been explored in Alberge et al. (2019) and Bar-Shalom et al. (2022), while neural network activation graph features have been studied in, *e.g.*, Lacombe et al. (2021) and Horta et al. (2021). The related issue of the robustness of various algorithms to diverse types of shifts has been recently investigated in Wiles et al. (2021a).

## 3 BACKGROUND

### 3.1 SHIFT DETECTION WITH TWO-SAMPLE TESTS

There can often be a shift between the distribution $\mathbb{P}_0$ of data on which a model has been trained and tested and the distribution $\mathbb{P}_1$ of the data on which it is used after deployment; many factors can cause such a shift, *e.g.*, a change in the environment, in the data acquisition process, or the training set being unrepresentative. Detecting shifts is crucial to understanding, and possibly correcting, potential losses in performance; even shifts that do not strongly impact accuracy can be important symptoms of inaccurate assumptions or changes in deployment conditions.

Additional assumptions can sometimes be made on the nature of the shift. In the context of a classification task, where data points are of the shape $(x, y)$ with $x$ the feature vector and $y$ the label,

a shift that preserves the conditional distribution $p(x|y)$ (but allows the proportion of each label to vary) is called *label shift*. Conversely, a *covariate shift* occurs when $p(y|x)$ is preserved, but the distribution of $p(x)$ is allowed to change. In this article, we focus on the arguably harder case of covariate shifts. See Section 5 for examples of such shifts in numerical experiments.

Shifts can be detected using *two-sample tests*: that is, a statistical test that aims at deciding between the two hypotheses

$$H_0 : \mathbb{P}_0 = \mathbb{P}_1 \text{ and } H_1 : \mathbb{P}_0 \neq \mathbb{P}_1,$$

given two random sets of samples, $X_0$ and $X_1$, independently drawn from two distributions $\mathbb{P}_0$ and $\mathbb{P}_1$. To do so, many statistics have been derived, depending on the assumptions made on $\mathbb{P}_0$ and $\mathbb{P}_1$. In the case of distributions supported on $\mathbb{R}$, one such test is the *univariate Kolmogorov-Smirnov (KS) test*, of which we make use in this article. Given, as above, two sets of samples $X_0, X_1 \subset \mathbb{R}$, consider the empirical distribution functions $F_i(z) := \frac{1}{\mathrm{Card}(X_i)} \sum_{x \in X_i} 1_{x \leq z}$ for $i = 0, 1$ and $z \in \mathbb{R}$. Then the statistic associated with the KS test and the samples is $T := \sup_{z \in \mathbb{R}} |F_0(z) - F_1(z)|$. If $\mathbb{P}_0 = \mathbb{P}_1$, the distribution of $T$ is independent of $\mathbb{P}_0$ and converges to a known distribution when the sizes of the samples tend to infinity (under mild assumptions) Smirnov (1939). Hence approximate $p$-values can be derived. The KS test can also be used to compare multivariate distributions: if $\mathbb{P}_0$ and $\mathbb{P}_1$ are distributions on $\mathbb{R}^D$, a $p$-value $p_i$ can be computed from the samples by comparing the $i$-th entries of the vectors of $X_0, X_1 \subset \mathbb{R}^D$ using the univariate KS test, for $i = 1, \ldots, D$. A standard and conservative way of combining those $p$-values is to reject $H_0$ if $\min(p_1, \ldots, p_D) < \alpha/D$, where $\alpha$ is the significance level of the test. This is known as the *Bonferroni correction* Voss & George (1995). Other tests tackle the multidimensionality of the problem more directly, such as the *Maximum Mean Discrepancy (MMD) test*, though not necessarily with greater success (see, *e.g.*, Ramdas et al. (2014)).

### 3.2 Neural Networks

We now recall the basics of *neural networks* (NN), which will be our main object of study. We define a neural network[1] as a (finite) sequence of functions called *layers* $f_1, \ldots, f_L$ of the form $f_\ell \colon \mathbb{R}^{n_\ell} \to \mathbb{R}^{n_{\ell+1}}, x \mapsto \sigma_\ell(W_\ell \cdot x + b_\ell)$, where the parameters $W_\ell \in \mathbb{R}^{n_{\ell+1} \times n_\ell}$ and $b_\ell \in \mathbb{R}^{n_{\ell+1}}$ are called the weight matrix and the bias vector respectively, and $\sigma_\ell$ is an (element-wise) activation map (*e.g.*, sigmoid or ReLU). The neural network encodes a map $F \colon \mathbb{R}^d \to \mathbb{R}^D$ given by $F = f_L \circ \cdots \circ f_1$. We sometimes use $F$ to refer to the neural network as a whole, though it has more structure.

When the neural network is used as a classifier, the last activation function $\sigma_L$ is often taken to be the *softmax* function, so that $F(x)_i$ can be interpreted as the confidence that the network has in $x$ belonging to the $i$-th class, for $i = 1, \ldots, D$. For this reason, we use the terminology *confidence vector* (CV) for the output $F(x) \in \mathbb{R}^D$. The true class of $x$ is represented by a label $y = (0, \ldots, 0, 1, 0, \ldots, 0) \in \mathbb{R}^D$ that takes value 1 at the coordinate indicating the correct class and 0 elsewhere. The parameters of each layer $(W_\ell, b_\ell)$ are typically learned from a collection of training observations and labels $\{(x_n, y_n)\}_{n=1}^N$ by minimizing a cross-entropy loss through gradient descent, in order to make $F(x_n)$ as close to $y_n$ as possible on average over the training set. The *prediction* of the network on a new observation $x$ is then given by $\arg\max_{i=1,\ldots,D} F(x)_i$, and its (test) *accuracy* is the proportion of correct predictions on a new set of observations $\{(x'_n, y'_n)\}_{n=1}^{N'}$, that is assumed to have been independently drawn from the same distribution as the training observations. In this work, we consider NN classifiers that have already been trained on some training data and that achieve reasonable accuracies on test data following the same distribution as training data.

### 3.3 Activation Graphs

Given an instance $x = x_0 \in \mathbb{R}^d$, a trained neural network $f_1, \ldots, f_L$ with $x_{\ell+1} = f_\ell(x_\ell) = \sigma_\ell(W_\ell \cdot x_\ell + b_\ell)$ and a layer $f_\ell \colon \mathbb{R}^{n_\ell} \longrightarrow \mathbb{R}^{n_{\ell+1}}$, we can define a weighted graph, called the *activation graph* $G_\ell(x)$ of $x$ for the layer $f_\ell$, as follows. We let $V := V_\ell \sqcup V_{\ell+1}$ for two sets $V_\ell$ and $V_{\ell+1}$ of cardinality $n_\ell$ and $n_{\ell+1}$ respectively. The edges are defined as $E := V_\ell \times V_{\ell+1}$. To each edge $(i, j) \in E_\ell$, we associate the weight $w_{i,j}(x) := W_\ell(j, i) \cdot x_\ell(i)$, where $x_\ell(i)$ (resp. $W_\ell(j, i)$) denotes the $i$-th coordinate of $x_\ell \in \mathbb{R}^{n_\ell}$ (resp. entry $(j, i)$ of $W_\ell \in \mathbb{R}^{n_{\ell+1} \times n_\ell}$). The activation

---

[1]While our exposition is restricted to sequential neural networks for the sake of concision, our representations are well-defined for other types of neural nets (*e.g.*, recurrent neural nets).

graph $G_\ell(x)$ is the weighted graph $(V, E, \{w_{i,j}(x)\})$, which can be conveniently represented as a $n_\ell \times n_{\ell+1}$ matrix whose entry $(i, j)$ is $w_{i,j}(x)$. Intuitively, these activation graphs—first considered in Gebhart et al. (2019)—represent how the network "reacts" to a given observation $x$ at inner-level, rather than only considering the network output (*i.e.*, the Confidence Vector).

## 4 TWO-SAMPLE STATISTICAL TESTS USING MAGDiff

### 4.1 THE MAGDiff REPRESENTATIONS

Let $\mathbb{P}_0$ and $\mathbb{P}_1$ be two distributions for which we want to test $H_0 : \mathbb{P}_0 = \mathbb{P}_1$. As mentioned above, two-sample statistical tests tend to underperform when used directly on high-dimensional data. It is thus common practice to extract lower-dimensional representations $\Psi(x)$ from the data $x \sim \mathbb{P}_i$, where $\Psi \colon \mathrm{supp}\, \mathbb{P}_0 \cup \mathrm{supp}\, \mathbb{P}_1 \to \mathbb{R}^N$. Given a classification task with classes $1, \ldots, D$, we define a family of such representations as follows. Let $T \colon \mathrm{supp}\, \mathbb{P}_0 \cup \mathrm{supp}\, \mathbb{P}_1 \to V$ be any map whose codomain $V$ is a Banach space with norm $\|\cdot\|_V$. For each class $i \in \{1, \ldots, D\}$, let $\mathbb{P}_{0,i}$ be the conditional distribution of data points from $\mathbb{P}_0$ in class $i$. We define

$$\Psi_i(x) \coloneqq \|T(x) - \mathbb{E}_{\mathbb{P}_{0,i}}[T(x')]\|_V$$

for $x \in \mathrm{supp}\, \mathbb{P}_0 \cup \mathrm{supp}\, \mathbb{P}_1$. Given a fixed finite dataset $x_1, \ldots, x_m \overset{\mathrm{iid}}{\sim} \mathbb{P}_0$, we similarly define the approximation

$$\tilde{\Psi}_i(x) \coloneqq \|T(x) - \frac{1}{m_i} \sum_{j=1}^{m_i} T(x_j^i)\|_V,$$

where $x_1^i, \ldots, x_{m_i}^i$ are the points whose class is $i$. This defines a map $\tilde{\Psi} \colon \mathrm{supp}\, \mathbb{P}_0 \cup \mathrm{supp}\, \mathbb{P}_1 \to \mathbb{R}^D$.

The map $T \colon \mathrm{supp}\, \mathbb{P}_0 \cup \mathrm{supp}\, \mathbb{P}_1 \to V$ could *a priori* take many shapes. In this article, we assume that we are provided with a neural network $F$ that has been trained for the classifying task at hand, as well as a training set drawn from $\mathbb{P}_0$. We let $T$ be the activation graph $G_\ell$ of the layer $f_\ell$ of $F$ represented as a matrix, so that the expected values $\mathbb{E}_{\mathbb{P}_{0,i}}[G_\ell(x')]$ (for $i = 1, \ldots, D$) are simply mean matrices, and the norm $\|\cdot\|_V$ is the Frobenius norm $\|\cdot\|_2$. We call the resulting $D$-dimensional representation *Mean Activation Graph Difference* (MAGDiff):

$$\mathtt{MAGDiff}(x)_i \coloneqq \|G_\ell(x) - \frac{1}{m_i} \sum_{j=1}^{m_i} G_\ell(x_j^i)\|_2,$$

for $i = 1, \ldots, D$, where $x_1^i, \ldots, x_{m_i}^i$ are, as above, samples of the training set whose class is $i$. Therefore, for a given new observation $x$, we derive a vector $\mathtt{MAGDiff}(x) \in \mathbb{R}^D$ whose $i$-th coordinate indicates whether $x$ activates the chosen layer of the network in a similar way "as training observations of the class $i$".

Many variations are possible within that framework. One could, *e.g.*, consider the activation graph of several consecutive layers, use another matrix norm, or apply Topological Data Analysis techniques to compute a more compact representation of the graphs, such as the *topological uncertainty* Lacombe et al. (2021). In this work, we focus on MAGDiff for dense layers, though it could be extended to other types.

### 4.2 COMPARISON OF DISTRIBUTIONS OF FEATURES WITH MULTIPLE KS TESTS

Given as above a (relatively low-dimensional) representation $\Psi \colon \mathrm{supp}\, \mathbb{P}_0 \cup \mathrm{supp}\, \mathbb{P}_1 \to \mathbb{R}^N$ and samples $x_1, \ldots, x_n \overset{\mathrm{iid}}{\sim} \mathbb{P}_0$ and $x_1', \ldots, x_m' \overset{\mathrm{iid}}{\sim} \mathbb{P}_1$, one can apply multiple univariate (coordinate-wise) KS tests with Bonferroni correction to the sets $\Psi(x_1), \ldots, \Psi(x_n)$ and $\Psi(x_1'), \ldots, \Psi(x_m')$, as described in Section 3. If $\Psi$ is well-chosen, a difference between the distributions $\mathbb{P}_0$ and $\mathbb{P}_1$ (hard to test directly due to the dimensionality of the data) will translate to a difference between the distributions $\Psi(x)$ and $\Psi(x')$ for $x \sim \mathbb{P}_0$ and $x' \sim \mathbb{P}_1$ respectively. Detecting such a difference serves as a proxy for testing $H_0 : \mathbb{P}_0 = \mathbb{P}_1$. In our experiments, we apply this procedure to the MAGDiff representations defined above (see Section 5.1 for a step-by-step description). This is a reasonable approach, as it is a simple fact that a generic shift in the distribution of the random variable $x \sim \mathbb{P}_0$ will in turn induce a shift in the distribution of $\Psi(x)$, as long as $\Psi$ is not constant[2];

---

[2]See the Supplementary Material, Section 3 for an elementary proof.

however, this does not give us any true guarantee, as it does not provide any quantitative result regarding the shift in the distribution of $\Psi(x)$. Such results are beyond the scope of this paper, in which we focus on the good experimental performance of the `MAGDiff` statistic.

### 4.3 DIFFERENCES FROM BBSD AND MOTIVATIONS

The BBSD method described in Lipton et al. (2018) and Rabanser et al. (2019) is defined in a similar manner, except that the representations $\Psi$ on which the multiple univariate KS tests are applied are simply the Confidence Vectors (CV) $F(x) \in \mathbb{R}^D$ of the neural network $F$ (or of any other classifier that outputs confidence vectors), rather than our newly proposed `MAGDiff` representations. In other words, they detect shifts in the distribution of the inputs by testing for shifts in the distribution of the outputs of a given classifier[3].

Both our method and theirs share advantages: the features are task- and data-driven, as they are derived from a classifier that was trained for the specific task at hand. They do not require the design or the training of an additional model specifically geared towards shift detection, and they have favorable algorithmic complexity, especially compared to some kernel-based methods. In particular, combining the KS tests with the Bonferroni correction spares us from having to calibrate our statistical tests with a permutation test, which can be costly as shown in Rabanser et al. (2019). A common downside is that the Bonferroni correction can be overly conservative; other tests might offer higher power. The main focus of this article is the relevance of the `MAGDiff` representations, rather than the statistical tests that we apply to them, and it has been shown in Rabanser et al. (2019) that KS tests yield state-of-the-art performances; as such, we did not investigate alternatives, though additional efforts in that direction might produce even better results.

The nature of the construction of the `MAGDiff` representations is geared towards shift detection since it is directly based on encoding differences (*i.e.*, deviations) from the mean activation graphs (of $\mathbb{P}_0$). Moreover, they are based on representations from deeper within the NN, which are less compressed than the CV - passing through each layer leads to a potential loss of information. Hence, we can hope for the `MAGDiff` to encode more information from the input data than the CV representations used in Rabanser et al. (2019) which focus on the class to which a sample belongs to, while sharing the same favorable dimensionality reduction properties. Therefore, we expect `MAGDiff` to perform particularly well with covariate shifts, where shifts in the distribution of the data do not necessarily translate to strong shifts in the distribution of the CV. Conversely, we do not hope for our representations to bring significant improvements over CV in the specific case of label shifts; all the information relative to labels available to the network is, in a sense, best summarized in the CV, as this is the main task of the NN. These expectations were confirmed in our experiments.

## 5 EXPERIMENTS

This experimental section is devoted to showcasing the use of the `MAGDiff` representations and its benefits over the well-established baseline CV when it comes to performing covariate shift detection. As detailed in Section 5.1, we combine coordinate-wise KS tests for both these representations. Note that in the case of CV, this corresponds exactly to the method termed *BBSDs* in Rabanser et al. (2019). Our code is provided in the Supplementary Material, as well as a more thorough presentation of the datasets and parameters used.

### 5.1 EXPERIMENTAL SETTINGS

**Datasets.** We consider the standard datasets MNIST LeCun et al. (1998), FashionMNIST (FM-NIST) Xiao et al. (2017), CIFAR-10 Krizhevsky & Hinton (2009), SVHN Netzer et al. (2011), as well as a lighter version of ImageNet (restricted to 10 classes) called Imagenette Ima.

**Architectures.** For MNIST and FMNIST, we used a simple CNN architecture consisting of 3 convolutional layers followed by 4 dense layers. For CIFAR-10 and SVHN, we considered (a slight modification, to account for input images of size $32 \times 32$, of) the ResNet18 architecture He et al.

---

[3]This corresponds to the best-performing variant of their method, denoted as *BBSDs* (as opposed to, *e.g.*, *BB-SDh*) in Rabanser et al. (2019).

(2015). For Imagenette, we used a pretrained ResNet18 model provided by Pytorch Res. With these architectures, we reached a test accuracy of $98.6\%$ on MNIST, $91.1\%$ on FMNIST, $94.1\%$ on SVHN, $81\%$ on CIFAR-10 and $99.2\%$ for Imagenette, validating the "well-trained" assumption mentioned in Section 4. Note that we used simple architectures, without requiring the networks to achieve state-of-the-art accuracy.

**Shifts.** We applied three types of shift to our datasets: Gaussian noise (additive white noise), Gaussian blur (convolution by a Gaussian distribution), and Image shift (random combination of rotation, translation, zoom and shear), for six different levels of increasing intensities (denoted by I, II,..., VI), and a fraction of shifted data $\delta \in \{0.25, 0.5, 1.0\}$. For each dataset and shift type, we chose the shift intensities in such a manner that the shift detection for the lowest intensities and low $\delta$ is almost indetectable for both methods (`MAGDiff` and CV), and very easily detectable for high intensities and values of $\delta$. Details (including the impact of the shifts on model accuracy) and illustrations can be found in the Supplementary Material.

**Sample size.** We ran the shift detection tests with sample sizes[4] $\{10, 20, 50, 100, 200, 500, 1000\}$ to assess how many samples a given method requires to reliably detect a distribution shift. A good method should be able to detect a shift with as few samples as possible.

**Experimental protocol.** In all of the experiments below, we start with a neural network that is pre-trained on the training set of a given dataset. The test set will be referred to as the *clean set* (CS). We then apply the selected shift (type, intensity, and proportion $\delta$) to the clean set and call the resulting set the *shifted set SS*; it represents the target distribution $\mathbb{P}_1$ in the case where $\mathbb{P}_1 \neq \mathbb{P}_0$.

As explained in Section 4, for each of the classes $i = 1, \ldots, D$ (for all of our datasets, $D = 10$), we compute the mean activation graph of a chosen dense layer $f_\ell$ of (a random subset of size 1000 of all) samples in the training set whose class is $i$; this yields $D$ mean activation graphs $G_1, \ldots, G_D$. We compute for each sample $x$ in $CS$ and each sample in $SS$ the representation $\texttt{MAGDiff}(x)$, where $\texttt{MAGDiff}(x)_i = \|G_\ell(x) - G_i\|_2$ for $i = 1, \ldots, D$ and $G_\ell(x)$ is the activation graph of $x$ for the layer $f_\ell$ (as explained in Section 4). Doing so, we obtain two sets $\{\texttt{MAGDiff}(x) \mid x \in CS\}$ and $\{\texttt{MAGDiff}(x') \mid x' \in SS\}$ of $D$-dimensional features with the same cardinality as the test set.

Now, we estimate the power of the test for a given sample size[5] $m$ and for a type I error of at most $0.05$; in other words, the probability that the test rejects $H_0$ when $H_1$ is true and when it has access to only $m$ samples from the respective datasets, and under the constraint that it does not falsely reject $H_0$ in more than $5\%$ of cases. To do so, we randomly sample (with replacement) $m$ elements $x'_1, \ldots, x'_m$ from $SS$, and consider for each class $i = 1, \ldots, D$ the discrete empirical univariate distribution $q_i$ of the values $\texttt{MAGDiff}(x'_1)_i, \ldots, \texttt{MAGDiff}(x'_m)_i$. Similarly, by randomly sampling $m$ elements from $CS$, we obtain another discrete univariate distribution $p_i$ (see Figure 1 for an illustration). Then, for each $i = 1, \ldots, D$, the KS test is used to compare $p_i$ and $q_i$ to obtain a $p$-value $\lambda_i$, and reject $H_0$ if $\min(\lambda_1, \ldots, \lambda_D) < \alpha/D$, where $\alpha$ is the threshold for the univariate KS test at confidence $0.05$ (*cf.* Section 3.1). Following standard bootstrapping protocol, we repeat that experiment (independently sampling $m$ points from $CS$ and $SS$, computing $p$-values, and possibly rejecting $H_0$) 1500 times; the percentage of rejection of $H_0$ is the estimated *power* of the statistical test (since $H_0$ is false in this scenario). We use the asymptotic normal distribution of the standard Central Limit Theorem to compute approximate $95\%$-confidence intervals on our estimate.

To illustrate that the test is well calibrated, we repeat the same procedure while sampling twice $m$ elements from $CS$ (rather than $m$ elements from $SS$ and $m$ elements from $CS$), which allows us to estimate the type I error (*i.e.*, the percentage of incorrect rejections of $H_0$) and assert that it remains below the significance level of $5\%$ (see, *e.g.*, Figure 2).

We experimented with a few variants of the `MAGDiff` representations: we tried reordering the coordinates of each vector $\texttt{MAGDiff}(x) \in \mathbb{R}^D$ in increasing order of the value of the associated confidence vectors. We also tried replacing the matrix norm of the difference to the mean activation graph by either its Topological Uncertainty (TU) Lacombe et al. (2021), or variants thereof. Early analysis suggested that these variations did not bring increased performances, despite their

---

[4]That is, the number of elements from the clean and shifted sets on which the statistical tests are performed; see the paragraph **Experimental protocol** for more details.

[5]The same sample size that is mentioned in the **Sample size** paragraph.

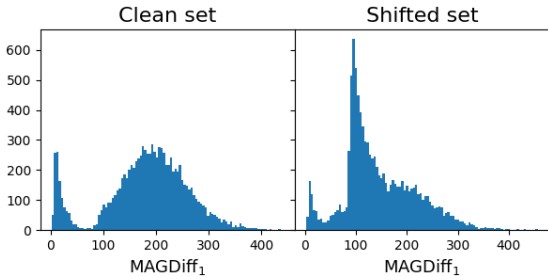

Figure 1: Empirical distributions of `MAGDiff_1` for the $10,000$ samples of the clean and shifted sets (MNIST, Gaussian noise, $\delta = 0.5$, last dense layer). For the clean set, the distribution of the component `MAGDiff_1` of `MAGDiff` exhibits a peak close to 0. This corresponds to those samples whose distance to the mean activation graph of (training) samples belonging to the associated class is very small, *i.e.*, these are samples that presumably belong to the same class as well. Note that, for the shifted set, this peak close to 0 is substantially diminished, which indicates that the activation graph of samples affected by the shift is no longer as close to the mean activation graph of their true class.

increased complexity. Experiments also suggested that `MAGDiff` representations brought no improvement over CV in the case of label shift. We also tried to combine (*i.e.*, concatenate) the CV and `MAGDiff` representations, but the results were unimpressive, which we attribute to the Bonferroni correction getting more conservative the higher the dimension. We thus only report the results for the standard `MAGDiff`.

**Competitor.** We used multiple univariate KS tests applied to CV (the method BBSDs from Rabanser et al. (2019)) as the baseline, which we denote by "CV" in the figures and tables, in contrast to our method denoted by "`MAGDiff`". The similarity in the statistical testing between BBSDs and `MAGDiff` allows us to easily assess the relevance of the `MAGDiff` features. We chose them as our sole competitors as it has been convincingly shown in Rabanser et al. (2019) that they outperform on average all other standard methods, including the use of dedicated dimensionality reduction models, such as autoencoders, or of multivariate kernel tests. Many of these methods are also either computationally more costly (to the point where they cannot be practically applied to more than a thousand samples) or harder to implement (as they require an additional neural network to be implemented) than both BBSDs and `MAGDiff`.

## 5.2 EXPERIMENTAL RESULTS AND INFLUENCE OF PARAMETERS.

We now showcase the power of shift detection using our `MAGDiff` representations in various settings and compare it to the state-of-the-art competitor CV. Since there were a large number of hyperparameters in our experiments (datasets, shift types, shift intensities, etc.), we started with a standard set of hyper-parameters that yielded representative and informative results according to our observations (MNIST and Gaussian noise, as in Rabanser et al. (2019), $\delta = 0.5$, sample size 100, `MAGDiff` computed with the last layer of the network) and let some of them vary in the successive experiments. We focus on the well-known MNIST dataset to allow for easy comparisons, and refer to the Supplementary Material for additional experimental results that confirm our findings on other datasets.

**Sample size.** The first experiment consists of estimating the power of the shift detection test as a function of the sample size (a common way of measuring the performance of such a test) using either the `MAGDiff` or the baseline CV representations. Figure 2 shows the powers of the KS tests using the `MAGDiff` (red curve) and CV (green curve) representations with respect to the sample size for the MNIST dataset. Here, we choose to showcase the results for Gaussian noise of intensities II, IV and IV with shift proportion $\delta = 0.5$.

It can clearly be seen that `MAGDiff` consistently and significantly outperformed the CV representations. While in both cases, the tests achieved a power of 1.0 for large sample sizes ($m \approx 1000$) and/or high shift intensity (VI), `MAGDiff` was capable of detecting the shift even with much lower sample sizes. This was particularly striking for the low intensity level II, where the test with CV was completely unable to detect the shift, even with the largest sample size, while `MAGDiff` was capable

**Power as a Function of Sample Size**

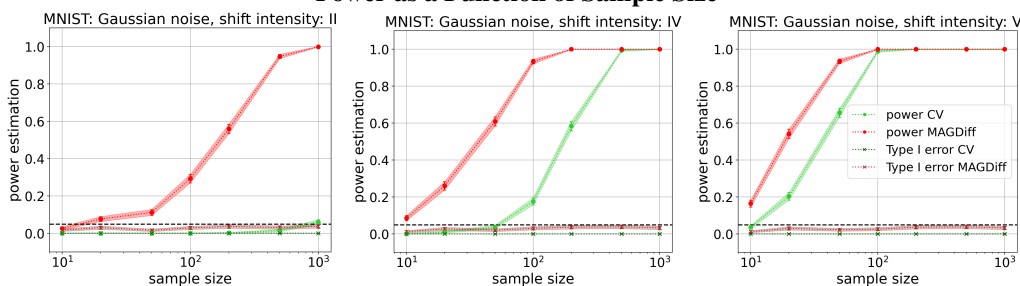

Figure 2: Power and type I error of the statistical test with `MAGDiff` (red) and CV (green) representations w.r.t. sample size (on a log-scale) for three different shift intensities (II, IV, VI) and fixed $\delta = 0.5$ for the MNIST dataset, Gaussian noise and last layer of the network, with estimated $95\%$-confidence intervals.

of reaching non-trivial power already for a medium sample size of $100$ and exceptional power for large sample size. Note that the tests were always well-calibrated. That is, the type I error remained below the significance level of $0.05$, indicated by the horizontal dashed black line in the figures.

To further support our claim that `MAGDiff` outperforms CV on average in other scenarios, we provide, in Table 1, averaged results over all parameters except the sample size. Though the precise values obtained are not particularly informative (due to the aggregation over very different sets of hyper-parameters), the comparison between the two rows remains relevant. In the Supplementary Material, a more comprehensive experimental report (including, in particular, the CIFAR-10 and Imagenette datasets) further supports our claims.

| | Averaged power (%) | | | | | | |
|---|---|---|---|---|---|---|---|
| Sample size | 10 | 20 | 50 | 100 | 200 | 500 | 1000 |
| `MAGDiff` | 7.4 | 17.1 | 27.6 | 40.7 | 54.7 | 71.4 | 80.4 |
| CV | 4.0 | 9.8 | 15.6 | 24.7 | 35.3 | 49.7 | 59.2 |

Table 1: Averaged test power of `MAGDiff` and CV over all hyper-parameters except sample size (dataset, shift type, $\delta$, shift intensity). A $95\%$-confidence interval for the averaged powers has been estimated via bootstrapping and is, in all cases, strictly contained in a $\pm 0.1\%$ interval.

**Impact of Shift Intensity**

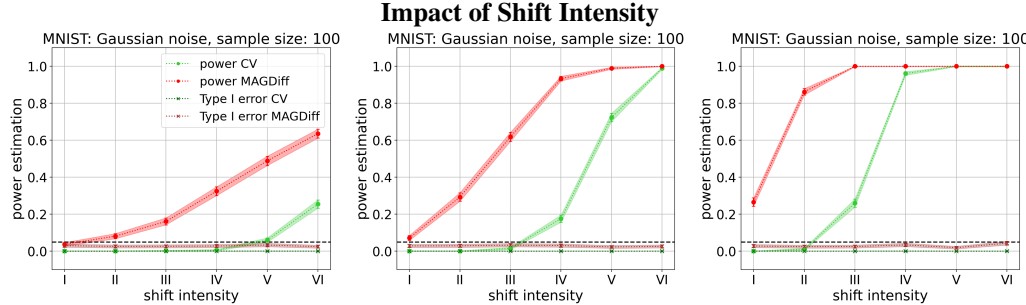

Figure 3: Power and type I error of the test with `MAGDiff` (red) and CV (green) features w.r.t. the shift intensity for Gaussian noise on the MNIST dataset with sample size $100$ and $\delta = 0.25$ (left), $\delta = 0.5$ (middle), $\delta = 1.0$ (right), for the last dense layer. The estimated $95\%$-confidence intervals are displayed around the curves.

**Shift intensity.** The first experiment suggests that `MAGDiff` representations perform particularly well when the shift is hard to detect. In the second experiment, we further investigate the influence of the shift intensity level and $\delta$ (which is, in a sense, another measure of shift intensity) on the power of the tests. We chose a fixed sample size of $100$, which was shown to make for a challenging yet doable task. The results in Figure 3 confirm that our representations were much more sensitive to weak shifts than the CV, with differences in power greater than $80\%$ for some intensities.

**Shift type.** The previous experiments focused on the case of Gaussian noise; in this experiment, we investigate whether the results hold for other shift types. As detailed in Table 2, the test with

**Impact of Shift Type**

| Shift | Feat. | Shift intensity | | | | | |
|---|---|---|---|---|---|---|---|
| | | Power of the test (%) | | | | | |
| | | I | II | III | IV | V | VI |
| GN | MD | $7.2 \pm 1.3$ | $29.3 \pm 2.3$ | $61.9 \pm 2.5$ | $93.3 \pm 1.3$ | $98.9 \pm 0.5$ | $100.0 - 0.0$ |
| | CV | $0.0 + 0.2$ | $0.1 \pm 0.1$ | $1.5 \pm 0.6$ | $17.6 \pm 1.9$ | $72.3 \pm 2.3$ | $98.9 \pm 0.5$ |
| GB | MD | $3.7 \pm 1.0$ | $4.3 \pm 1.0$ | $27.7 \pm 2.3$ | $63.1 \pm 2.4$ | $85.0 \pm 1.8$ | $92.4 \pm 1.3$ |
| | CV | $0.0 + 0.2$ | $0.0 + 0.2$ | $0.0 + 0.2$ | $0.4 \pm 0.3$ | $1.3 \pm 0.6$ | $5.3 \pm 1.1$ |
| IS | MD | $10.3 \pm 1.5$ | $32.7 \pm 2.4$ | $53.5 \pm 2.5$ | $78.5 \pm 2.1$ | $90.6 \pm 1.5$ | $98.9 \pm 0.5$ |
| | CV | $0.0 + 0.2$ | $0.1 \pm 0.2$ | $2.1 \pm 0.7$ | $11.5 \pm 1.6$ | $37.0 \pm 2.4$ | $86.3 \pm 1.7$ |

Table 2: Power of the two methods (our method, denoted as `MD`, and CV) as a function of the shift intensity for the shift types Gaussian noise (GN), Gaussian blur (GB) and Image shift (IS) on the MNIST dataset with $\delta = 0.5$, sample size 100, for the last dense layer. Red indicates that the estimated power is below $10\%$, blue that it is above $50\%$. The $95\%$-confidence intervals have been estimated as mentioned in Section 5.

`MAGDiff` representations reacted to the shifts even for low intensities of I, II, and III for all shift types (Gaussian blur being the most difficult case), while the KS test with CV was unable to detect anything. For medium to high intensities III, IV, V and VI, `MAGDiff` again significantly outperformed the baseline and reaches powers close to 1 for all shift types. For the Gaussian blur, the shift remained practically undetectable using CV.

**`MAGDiff` with respect to different layers.** The NN architect we used with MNIST and FMNIST consisted had several dense layers before the output. As a variation of our method, we investigate the effect on the shift detection when computing our `MAGDiff` representations with respect to different layers[6]. More precisely, we consider the last three dense layers denoted by $\ell_{-1}$, $\ell_{-2}$ and $\ell_{-3}$, ordered from the closest to the network output ($\ell_{-1}$) to the third from the end ($\ell_{-3}$). The averaged results over all parameters and noise types are in Table 3.

**Choice of Layer**

| Dataset | Features | | | |
|---|---|---|---|---|
| | Averaged power (%) | | | |
| | CV | $\ell_{-1}$ | $\ell_{-2}$ | $\ell_{-3}$ |
| MNIST | 25.1 | 51.9 | 53.0 | 56.4 |
| FMNIST | 46.2 | 44.9 | 47.6 | 53.7 |

Table 3: Averaged performance of the various layers for `MAGDiff` over all other parameters (for MNIST and FMNIST), compared to BBSD with CV. A $95\%$ confidence interval for the averaged powers was estimated and is in all cases contained in a $\pm 0.1\%$ interval.

In the case of MNIST we only observe a slight increase in power when considering layer $\ell_{-3}$ further from the output of the NN. In the case of FMNIST, on the other hand, we clearly see a much more pronounced improvement when switching from $\ell_{-1}$ to $\ell_{-3}$. This hints at the possibility that features derived from encodings further within the NN can, in some cases, be more pertinent to the task of shift detection than those closer to the output.

## 6 CONCLUSION

In this article, we derive new representations `MAGDiff` from the activation graphs of a trained NN classifier. We empirically show that using `MAGDiff` representations for data set shift detection via coordinate-wise KS tests (with Bonferroni correction) significantly outperforms the baseline given by using confidence vectors established in Lipton et al. (2018), while remaining equally fast and easy to implement, making `MAGDiff` representations an efficient tool for this critical task.

Our findings open many avenues for future investigations. We focused on classification of image data in this work, but our method is a general one and can be applied to other settings. Moreover, adapting our method to regression tasks as well as to settings where shifts occur gradually is feasible and a starting point for future work. Finally, exploring variants of the `MAGDiff` representations—considering several layers of the network at once, extending it to other types of layers, extracting finer topological information from the activation graphs, weighting the edges of the graph by back-propagating their contribution to the output, etc.—could also result in increased performance.

---

[6]Since ResNet18 only has a single dense layer after its convolutional layers, there is no choice to be made in the case of CIFAR-10, SVHN and Imagenette.

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
