# OpenReview forum: "MAGDiff: Covariate Data Set Shift Detection via Activation Graphs of Deep Neural Networks"
_ICLR.cc/2024/Conference — Submitted to ICLR 2024_

### Official Review · Reviewer_QE2P · 2023-10-27

**Soundness:** 3 good
**Presentation:** 3 good
**Contribution:** 2 fair
**Rating:** 5
**Confidence:** 3

**Summary:**

The authors propose a new type of representations for covariate shift detection, based on activation graphs of neural network layers. They consider classification tasks, where a pretrained, well-performing classifier is given. The Kolmogorov-Smirnov test is then performed on class-conditioned activation graphs of the original and target distributions. The authors compare the proposed method on several image datasets against a single (albeit state-of-the-art) baseline, and analyse robustness of the methods to changing conditions, e.g., sample size of different shift types.

**Strengths:**

- The paper is written in a clear and concise manner
- The proposed method outperforms the baseline
- Experimental details are included in the main manuscript

**Weaknesses:**

- Authors compare against just one baseline, while e.g., [1] could be applicable in this scenario as well
- The novelty of the method as compared to the baseline is rather limited

[1] Kirchler, Matthias, et al. "Two-sample testing using deep learning." *International Conference on Artificial Intelligence and Statistics*. PMLR, 2020.

**Questions:**

-

---

### Official Review · Reviewer_hyAB · 2023-10-31

**Soundness:** 2 fair
**Presentation:** 3 good
**Contribution:** 3 good
**Rating:** 5
**Confidence:** 3

**Summary:**

This paper proposes a new method called MAGDiff for detecting covariate shifts in data distributions by comparing activation graphs in neural networks. This method works in principle for any NN classifier with dense layers. By using this as a statistic and combining it with the methods proposed in prior work, the authors obtain a novel method for covariate detection. A comparison to the BBSD method which sets SOTA on covariate shift is done, and this method is seen to outperform. Code is provided for reproducibility.

**Strengths:**

To the best of my knowledge, the idea of using activation graphs for shift detection is novel. MAGDiff provides a way to leverage neural network representations for this task without retraining models. Detecting distribution shift is an important problem in deploying machine learning systems. The paper shows MAGDiff outperforms a strong baseline in a variety of scenarios. This appears to be a significant result. The paper is well-written and clearly explains the proposed method. Section 3 makes it accessible to someone with no strong background in the distribution shift literature and problem. The experiments systematically evaluate performance across datasets, shift types and intensities. I'm not fully convinced that the set of shifts applied are sufficiently strong be realistic in practice, but the comparison with SOTA CV detector for even these simple tasks is compelling. The limitations discussed in the experimental protocol section are thorough.

**Weaknesses:**

The main weakness is the limited number of datasets and architectures that this method is being tested on. MNIST, FMNIST, and CIFAR have by now become fairly easy and simple datasets. The use of Imagenette is appreciated (and the results there seem compelling), but a broader variety of empirical settings would drastically strengthen the paper.  As distribution shift is highly relevant for real-world deployment, more rigorous and diverse empirical analysis is needed to fully validate the claims in this paper. The current experiments are reasonable for an initial investigation but leave open questions about how MAGDiff would perform on more complex and realistic data. Further studies on more realistic datasets would likely lead me to raise my score.

There is also some lacking theoretical motivation for why this method should work. In the opinion of the reviewer, this is not as important as the first weakness listed.

**Questions:**

It would be interesting to see if this can be extended outside of the supervised setting, e.g. to autoregressive tasks.

Could you observe improved results by somehow combining the tests for different layers of a given network? In the case of the ConvNets, I suppose this requires you to develop the method for the convolutional layer.

---

### Official Review · Reviewer_WvRV · 2023-10-31

**Soundness:** 3 good
**Presentation:** 3 good
**Contribution:** 2 fair
**Rating:** 3
**Confidence:** 4

**Summary:**

This paper proposes the use of MAGDiff representation, extracted from the activation graph of a trained neural network classifier, to construct tests for covariate shift detection in datasets that are used to train and query the classifier. The paper shows that coordinate-wise two-sample Kolmogorov Smirnov (KS) tests for each class can be combined to create a covariate detector with higher power than the baseline (Black box shift detection, BBSD) that only relies on the output logits (Confidence Vectors) to construct the representation.

**Strengths:**

Originality and significance: The use of neural network internal representation (activation graph) to construct a covariate-shift detector is the main contribution of this work. Most of the rest of the paper closely follows the baseline BBSD method (Lipton et al., 2018). The empirical results are interesting and shed some light on the potential use of internal NN representation as proxies for constructing tests that measure changes in the input space.

Quality and clarity: The paper is easy to follow and the experiments are explained in sufficient detail.

**Weaknesses:**

The original results in this work are fairly limited and the paper defers a closer study of several key aspects of the proposed approach to future work. IMO, several key questions are not answered and the experiments are insufficient.

The paper suggests looking inside the model for representations with higher covariate shift detection power, but fails to pinpoint where exactly one should be looking inside the model. IMO, the study with respect to different layers (the last section of the empirical results) should have been the main focus of the paper, with more models and certainly more layers. As is, the experiment looks like an afterthought, and the optimal choice of layer is at the boundary of the experimental setup, suggesting that using earlier layers might have even higher detection power. Of course we don’t know if this is the case since the experimental setup is somewhat minimal. We also don’t know if these results extend beyond the toy examples of MNIST and FMNIST.

I don’t want to argue for more published baselines to be added to the paper (will leave that to the proverbial reviewer number 2). But IMO the paper must have at least included a trivial variant of their method as the true baseline: using the activations themselves instead of the activation graph as the internal representation, i.e. setting $T(x)=x$ in the derivations of section 4.1 (please number your equations). This is an extension of BBSD without involving layer weights and should be compared to MAGDiff (per layer), to gauge the effect of the linear map on top of the previous layer activations as it relates to covariate shift detection.

Even on the very limited toy models, the work does not cover how to combine representations across multiple layers, even though such a combination is likely to result in higher power tests. One could postulate that just concatenating all middle layer activations to form $T(x)$ might have results on par or better than those reported in the paper. But of course we don’t know if this is the case.

I found it very frustrating to read a paper with a good core idea that fails to answer so many key questions around the proposed method, excludes “unimpressive” results that could have shed some light on the choice of hyper parameters, and does virtually no ablation studies to gauge the importance of solution components (e.g. the use activations instead of activation graph).

**Questions:**

Some questions and minor notes:

- Did you attempt using the (hidden) activations instead of the activation graph?
- If I took a hidden activation vector and use a random linear projection of it (i.e. MAGDiff with a randomly chosen W), how does it compare to using the weights from the trained model? In other words, how much of the test power can be attributed to using a trained W?
- Say, you take a model trained on a dataset A (e.g. MNIST), and try to use the activation graph of this model to measure covariate shift of another dataset B of similar input shape (e.g. FMNIST). You won’t have the “well-trained” model assumption, but you might still have a good proxy for measuring covariate shift (lambda values need to be set w.r.t. dataset B). Have you tried using a model trained on one dataset to check for data-shift in another dataset?
- For many experiments the choice of layer is not specified. The text in the paper says “a chosen dense layer” without indicating which one. My guess is that $l_{-1}$ is used for all experiments (and referred to as MagDiff$_1$).
- Equations could use label/numbering.
- The citation style is incorrect in many places (wrong macro). When the reference does not read as part of the sentence it should be in parentheses.

---

### Official Review · Reviewer_f2j1 · 2023-10-31

**Soundness:** 3 good
**Presentation:** 3 good
**Contribution:** 2 fair
**Rating:** 5
**Confidence:** 3

**Summary:**

The paper, "MAGDIFF: Covariate Data Set Shift Detection via Activation Graphs of Neural Networks", addresses the challenge of neural networks' sensitivity to shifts in data distribution. Such shifts can significantly degrade the performance of machine learning models. To address this, the authors introduce a new family of representations termed MAGDiff. These representations are derived by comparing the activation graphs of a given neural network classifier for samples from the training distribution and target distribution. The MAGDiff approach enables efficient covariate data shift detection without necessitating a new model specifically for this task. By employing these representations with two-sample tests, MAGDiff has demonstrated enhanced performance in detecting data set shifts compared to baselines.

**Strengths:**

S1. The paper introduces a novel concept, MAGDiff, which focuses on comparing activation graphs for shift detection. This approach is distinct from one of the state-of-the-art baselines that only looks at context vectors.

S2. The empirical results presented in the paper show that the proposed method outperforms the state-of-the-art BBSD technique, showcasing the quality and robustness of MAGDiff. Given the ubiquity and increasing reliance on neural networks in real-world applications, the capability to efficiently detect data set shifts is crucial. MAGDiff's methodology has potential impact for ensuring neural network reliability.

S3. The paper provide a clear and coherent overview of the problem, the proposed solution, and its advantages.

**Weaknesses:**

W1. While MAGDiff showcases success in the datasets mentioned, understanding its performance across a wider variety of datasets would enhance its credibility.

W2. How the method deals with very deep neural networks, where activation graphs can be highly complex, is not immediately clear. Currently the method assumes only simple fully connected networks to generate activation graphs. However, to utilize the potential of the method the more common architectures such CNNs, RNNs, Transformers are overlooked. How the model handles residual connections are also not clear. This lack of extension to the aforementioned models limits the applicability of the method in practice and does not give a real-benefit over BBSD since in practice fully-connected layers are often used at the very end of the models only.

W3. A more detailed breakdown comparing MAGDiff with other more recent methods, particularly highlighting where it excels or falls short, would provide a clearer picture for readers. BBSD is from 2018 and in the past 5 years there has been many other techniques that deal with such dataset shifts (some highlighted already in the paper but not directly used as baseline). Overall the work needs more additional baselines to make the contribution stronger.

W4. The paper mentions that there is no gain in using their method over BBSD for label-shift. But the experiments in the main body of the paper forego showing this. Additional experiments specific to label shift and not just covariate should be included in the paper to improve its soundness.

W5. For MAGDiff to be applicable in real-world applications it needs to have access to (or be provided beforehand with) the initial dataset that the model is trained on (CS). This is a challenge in using the model to detect dataset shift in practice.

**Questions:**

Q1. How does MAGDiff handle neural networks with different architectures or activation functions, given that activation graphs might differ substantially?

Q2. Given the reliance on activation graphs, how does MAGDiff deal with potential noise or irregularities in these graphs?

Q3. Are there any limitations or scenarios where MAGDiff might not be the ideal choice for shift detection?

Q4. Could the authors provide more insights into the computational overhead introduced by MAGDiff, especially for larger neural networks?

Q5. How would MAGDiff adapt or be integrated into real-time applications where timely detection of shifts is essential?

Q6. Section 3.2, you mention “reasonable accuracies” as a requirement, what constitutes as reasonable for various tasks? Does the method break for low accuracies? One possible way to analyze this in your experiments would be to use relatively low performing models and conducting the same analyses as you did.

Q7. Need a figure to clear up how activation graphs are formed.

Q8. Make your math more explained in text (e.g. supp $P0$  U supp  $P1 \rightarrow R^N$). Additionally, what is $N$ for instance? (It is first referenced in 4.1).

Q9. The choice of norm is not justified so it would be beneficial to show in the appendix effect of at least two norms in calculating the test statistic.

Q10. How we does MAGDiff generalize to large datasets? The experiments does not consider cases with large number of classes and how the model performs in such cases. I understand that the authors are directly comparing with BBSD and some of their experiments, but while MNIST was acceptable as the highlighting dataset in 2018, it is not in 2023 and so very large classes would be where this model can show its superior performance and highlight its applicability. Also it could shed light to some limitations for instance the fact that MAGDiff is calculating metrics independently for each class how does scale with large classes and not just 10.

Q11. What happens if we have a large pronounced imbalance between our classes?

Minor Details:
MQ1. In section 3.1 “i” is used for feature indexing but in section 3.2 “i” is used for class index. Would suggest changing the second one to “k” for clarity. Additionally, often “D” is used for the feature set, so using a different indicator like “K” for classes would be better in the flow of your arguments.

---

### Author Response · Authors · 2023-11-22

Thank you very much to all the reviewers for their helpful comments and suggestions. We have read them with attention, though we might not have the time to implement all of them.

---

### Meta-Review · Area_Chair_xLu3 · 2023-12-15

**Metareview:**

The paper aims to address the challenge of neural networks' sensitivity to shifts in data distribution. Although this is an important problem to be addressed, the paper lacks in several aspects, such as clarity and comparison with more recent methods, which may need further development.

**Justification For Why Not Higher Score:**

The paper lacks in various aspects and may need more development.

**Justification For Why Not Lower Score:**

NA.

---

### Decision · Program_Chairs · 2024-01-16

Reject